# Non-Metastatic Esophageal Adenocarcinoma: Circulating Tumor Cells in the Course of Multimodal Tumor Treatment

**DOI:** 10.3390/cancers11030397

**Published:** 2019-03-21

**Authors:** Jasmina Kuvendjiska, Peter Bronsert, Verena Martini, Sven Lang, Martha B. Pitman, Jens Hoeppner, Birte Kulemann

**Affiliations:** 1Department of General and Visceral Surgery, Medical Center—University of Freiburg, 79106 Freiburg, Germany; jasmina.kuvendjiska@uniklinik-freiburg.de (J.K.); verena.martini@uniklinik-freiburg.de (V.M.); sven.lang@uniklinik-freiburg.de (S.L.); birte.kulemann@uniklinik-freiburg.de (B.K.); 2Medical Faculty, University of Freiburg, 79106 Freiburg, Germany; peter.bronsert@uniklinik-freiburg.de; 3Institute for Surgical Pathology, Medical Center—University of Freiburg, 79106 Freiburg, Germany; 4Department of Pathology & Andrew L. Warshaw, MD Institute for Pancreatic Cancer Research, Massachusetts General Hospital/Harvard Medical School, Boston, MA 02114, USA; mpitman@partners.org

**Keywords:** esophageal cancer, adenocarcinoma, circulating tumor cells, metastasis, chemotherapy, chemoradiation, surgery

## Abstract

Background: Isolation of circulating tumor cells (CTC) holds the promise to improve response-prediction and personalization of cancer treatment. In this study, we test a filtration device for CTC isolation in patients with non-metastatic esophageal adenocarcinoma (EAC) within recent multimodal treatment protocols. Methods: Peripheral blood specimens were drawn from EAC patients before and after neoadjuvant chemotherapy (FLOT)/chemoradiation (CROSS) as well as after surgery. Filtration using ScreenCell^®^ devices captured CTC for cytologic analysis. Giemsa-stained specimens were evaluated by a cytopathologist; the cut-off was 1 CTC/specimen (6 mL). Immunohistochemistry with epithelial (pan-CK) and mesenchymal markers (vimentin) was performed. Results: Morphologically diverse malignant CTCs were found in 12/20 patients in at least one blood specimen. CTCs were positive for both vimentin and pan-CK. More patients were CTC positive after neoadjuvant therapy (6/20 vs. 9/15) and CTCs per/ml increased in most of the CTC-positive patients. After surgery, 8/13 patients with available blood specimens were still CTC positive. In clinical follow-up, 5/9 patients who died were CTC-positive. Conclusions: Detection of CTC by filtration within multimodal treatment protocols of non-metastatic EAC is feasible. The rate of CTC positive findings and the quantity of CTCs changes in the course of multimodal neoadjuvant chemoradiation/chemotherapy and surgery.

## 1. Introduction

Esophageal adenocarcinoma (EAC) is one of the most rapidly increasing tumor entities in the Western world. Despite application of multimodal treatment protocols, the overall outcome is still limited, even in non-metastatic disease. Five-year survival rates are about 35% with surgery alone and 45% with neoadjuvant chemoradiation (NeoCRT) plus surgery in the “ChemoRadiotherapy for Oesophageal cancer followed by Surgery Study” (CROSS) trial [1]. More than 50% of all patients who undergo an apparently curative, complete resection of a clinically non-metastatic primary tumor will eventually relapse or develop distant metastases [1]. This relapse is supposed to be caused by clinically occult micro-metastases present in secondary organs at an early time-point. These metastases originate from tumor cells that escape from the primary tumor and subsequently circulate in the vascular system, which are the so-called circulating tumor cells (CTC). They have been found to be of clinical utility in predicting response to treatment and prognosis in several malignancies like breast and prostate cancer [1,2,3,4]. There are, however, limited reports about the significance of CTC in patients with gastrointestinal malignancy [5]. In a study about the significance of CTC in patients with diverse gastrointestinal cancers including 23 patients with metastatic esophageal cancer, about 21% (5/23) of these patients were observed to be CTC-positive and had a significantly shorter overall survival than CTC-negative patients [6]. In another observation, CTC were found in 20% of the patients using epithelial surface antigen (EpCAM)-dependent isolation devices in EAC. These patients showed significantly shorter relapse-free and overall survival in an obsolete surgery-alone treatment protocol. In addition, multivariate analysis identified the presence of CTC as a strong, independent prognostic marker of tumor recurrence and overall survival [7]. Technically-different methods for the detection of CTC have been developed, including immunocytochemistry, reverse transcription polymerase chain reaction, flow-cytometry, and isolation by size (filtration) [5,8,9,10].

There is emerging evidence that there are cells in transit that escape CTC enrichment methods based on epithelial surface antigens due to different, more mesenchymal surface antigens in a course of the epithelial-mesenchymal transition (EMT) [11,12]. During the process of EMT, the epithelial tumor cells lose their polarity, which results in invasion into the surrounding tissues and penetration through vessel walls [13]. Thus, alternative, technically simple, less costly, and commercially available methods such as isolation by size filtration techniques (e.g. SceenCell^®^, Paris, France) have been developed [14]. 

The aim of this study is to assess the longitudinal presence and gross quantity of CTCs during the time course of combined radio-chemotherapeutic/chemotherapeutic and surgical multimodal curative treatment of non-metastatic EAC.

## 2. Results

### 2.1. Patients

Twenty patients with non-metastatic EAC were included in the study (CTC samples: time-point 1: *n* = 20, time-point 2: *n* = 15, time-point 3: *n* = 13). Fifteen patients had neoadjuvant chemotherapy according to the FLOT protocol (PeriCTX). Five patients had neoadjuvant chemoradiation according to the CROSS protocol (NeoCRT). One patient committed suicide during the neoadjuvant treatment. Two patients did not undergo a surgery due to poor physical condition or progressive disease with distant metastasis. Overall, 15 patients underwent curative esophagectomy including regional lymphadenectomy. One patient did not undergo final tumor resection since liver metastases were diagnosed during the operation. One patient underwent surgery outside the country and was lost to follow-up. Furthermore, three blood samples were not processed due to technical difficulties (clogging, belated specimen handling). The majority of the study population was male (18/20) and the average age was 62.1 years (Table 1).

### 2.2. CTC Detection

Single CTCs or cluster CTCs were detected in 60% (12/20) of the patient population at a minimum of one time-point in the course of treatment. The healthy volunteer samples did not contain cells that appear like CTCs. The number of CTC-positive (>1 CTC/specimen) patients increased after the neoadjuvant part of treatment. Furthermore, 30% (6/20) of the patients were positive at diagnosis, 60% (9/15) after the neoadjuvant therapy, and 61.5% (08/13) after surgery. Six of 12 CTC-positive patients turned from negative to positive during the therapy. Additionally, with exception of two patients, we observed an increase in the number of CTCs per milliliter blood after neoadjuvant therapy in every CTC-positive patient (Table 2, Figure 1 and Figure 2). Most patients, however, also showed a decrease of CTC counts after surgery (Table 2, Figure 1 and Figure 2).

One patient (Table 2, patient 8) with a small tumor and complete response to PeriCTX was CTC negative at diagnosis. The patient developed CTC after chemotherapy and showed an additional CTC increase after surgery. He suffered from early tumor recurrence and died in the second year after multimodal therapy even though his tumor was small and had shown complete response to therapy. One other patient with a small tumor without nodal involvement (Table 2, patient 13) had a high count of CTC cluster and several single CTC at diagnosis. The cell counts dropped after NeoCRT and no clusters were evident after surgery. He is still recurrence-free after two-year follow-up. A third patient (Table 2, patient 18) had a small, nodal negative tumor and subtotal regression after NeoCRT. He showed CTC cluster and very few single CTC at diagnosis and an increase after neoadjuvant treatment. After surgery, only 0.3 single CTC/ml single CTC were found in the specimen with no CTC cluster. The patient is also recurrence-free after two-year follow-up. Two other patients, however (Table 2, patients 9 and 10), had more advanced tumors that were not resected, but had no CTC at diagnosis and/or only very few CTC after neoadjuvant treatment. Both patients died within the first year of follow-up. Additionally, patient 12 (Table 2) had very few CTC at diagnosis, a slight increase after NeoCRT and a drop after surgery. He, however, also died within the first year of follow-up. In summary, the number of CTC often—but not universally—reflected the tumors´ response to treatment. 

Furthermore, morphological diversity of the isolated CTCs was observed. The CTCs showed variations in size, nuclear-to-cytoplasm (N/C) ratio, and nucleus morphology. Some patients presented CTC clusters, which consisted of much smaller cells than the single CTCs (Figure 3).

The cells identified as CTCs by the May-Grünwald Giemsa staining were analyzed for the presence of epithelial (pan-CK) and mesenchymal (vim) markers. Interestingly, all CTCs were positive for vim and pan-CK. They showed, however, different intensity levels of the staining within one patient (Figure 4). Thus, a rationale intensity scoring was not performed.

### 2.3. Tissue Analysis

Tissue analysis was performed on the operative specimens of 15 patients. The patients showed an almost equal distribution between the post-neoadjuvant TNM tumor stages (Table 1). Only 3/13 (23%) of the tissue samples had positive expression of vimentin in the immunohistochemistry (weak labelling: 1, moderate: 1, strong: 1) (Table 2). 

### 2.4. Survival Analysis

We conducted a one-year and two-year follow-up. In this short follow-up, we observed tumor relapse in 9/18 (50%) patients (Table 2). There were seven cases of exclusively distant metastasis, one patient with locoregional and distant recurrence, and one with local recurrence only. The one-year and two-year overall survival rates were 12/18 (67%) and 9/18 (50%) respectively. Due to the small number of patients, no correlation analysis was performed between the patients with tumor relapse and the presence of CTC. It was observed, though, that the majority of patients with tumor-recurrence in the follow-up were CTC-positive during at least one time-point. Five of the nine patients with tumor-relapse were CTC-positive (patients 2, 3, 8, 10, 12), one was CTC-negative (patient 7), and three patients had incomplete follow-up (patients 4, 9, 20) (Table 2). Of the nine patients deceased at two-year follow-up, five had CTCs at some time-point (patient 2, 3, 8, 10, 12). Only one patient was CTC-negative at all three time-points (patient 7). The remaining three patients had incomplete follow-up (patients 4, 9, 20).

## 3. Discussion

CTC are thought to be living clones of the primary tumor in transit with the ability to set up metastasis in distant organs. They have been found to be of clinical utility in predicting response to treatment and to sharpen prognosis in several malignancies like breast and prostate cancer [1,2,3,10,15]. There are, however, very limited studies on the significance of CTC in patients with esophageal adenocarcinoma (EAC) [7]. Clear progress has been made in the therapy of non-metastatic EAC, especially by using multimodal treatment protocols. To date, two multimodal treatment protocols, perioperative chemotherapy (PeriCTX,) according to the FLOT regimen [16], and neoadjuvant chemoradiation (NeoCRT), according to the CROSS regimen [1], are the international standards of care. Nevertheless, the prognosis is still impaired by a high rate of especially distant tumor recurrence [17]. 

In some studies, on non-metastatic breast cancer, it was shown that CTC could be found in almost all patients [18]. Under neoadjuvant chemotherapy, the number of peripherally circulating, epithelial tumor-suspicious cells dropped [18], increased again with continuing dissolution of the primary tumor, and persisted until the operation [19]. The decrease of these cells after neoadjuvant treatment had a high prediction level for recurrence-free survival after multimodal treatment of breast cancer patients [20]. Analogous to these results in breast cancer treatment, CTC could be a valuable staging tool for non-metastatic EAC to stratify patients into defined prognostic subgroups. An appropriate staging system is essential for determining treatment strategies, especially those involving neoadjuvant treatment in EAC patients. Despite the availability of several preoperative diagnostic techniques, accurate pretreatment staging remains inconsistent. Therefore, a novel tool for early tumor detection, adequate prognostic staging, and accurate therapy monitoring in EAC is urgently needed. The present study was conducted as a first step to assess the gross quantity and longitudinal presence of CTCs during combined radiochemotherapeutic or chemotherapeutic and consecutive surgical resection of non-metastatic EAC. The present report is the first study to investigate the presence of CTC in the course of current multimodal protocols in non-metastatic EAC. 

We found a variety of CTC types. Small cell clusters and medium to large single CTCs were the main CTC subgroups. Several patients showed CTC clusters, which are currently of high interest due to CTC-granulocyte interaction and the suspected induction of CTC proliferation by white blood cells [21]. The CTC clusters in this study were pure CTC clusters. Only a few clusters showed single eosinophilic granulocytes on—or close to—the cluster. New devices for isolation of CTC clusters are also currently being developed [22].

The largest study on single CTC in patients with esophageal cancer preoperatively investigated the presence of CTC in patients with EAC and squamous cell esophageal carcinoma, using the cell surface epithelial marker-based dependent detection method by CellSearch^®^. It reported a significant correlation of the presence of CTC with shorter relapse-free-survival in patients treated by surgery alone [7].

It is suspected that cells undergoing epithelial-to-mesenchymal transition (EMT), including loss of epithelial markers, can escape detection by epithelial marker-based detection systems. The results of a recent study on 21 patients with esophageal squamous cell carcinoma indicate that most CTCs from esophageal cancer had undergone EMT, since only 8.5% of the CTCs were epithelial-CTCs, 58.9% epithelial-mesenchymal-mixed CTCs, and 32.6% were mesenchymal CTCs [23]. In our study, all detected CTCs in EAC were of the epithelial-mesenchymal-mixed type with different intensity of the expression of vimentin and Cytokeratin. Due to a high staining background and autofluorescence of filter pores, a scored staining was not possible. CTC detection by CellSearch^®^ is solely based on antibodies against the epithelial EpCAM antigen. There is, thus, a possibility that a relevant subpopulation of CTCs is left undetected.

In light of the above, an antigen-independent size-based filtration method (ScreenCell^®^) was used in the present study and tested for feasibility in detecting CTCs in patients with EAC. We detected the presence of morphologically diverse single CTCs and CTC clusters. Since the patients in this study were in a non-metastatic state, the lowest CTC cut-off of 1 cell per sample was applied. Comparable to the results of observations made in patients treated with surgery alone, 30% of the patients with EAC were pretherapeutically CTC-positive in the present study [7]. Additionally, the number of CTCs increased after neoadjuvant therapy and decreased after surgery. This could be a result of dissolution of the primary tumor due to (radio-) chemotherapy. Due to the limited sample size, meaningful correlations with patient survival rates could not be tested in this first study. In a comparable study with 90 patients with esophageal squamous cell carcinoma, the presence of CTCs was examined before and after chemotherapy/chemoradiotherapy. The CTC status changed from negative to positive after therapy in approximately 13% of the patients, and 66.7% of these patients presented progressive disease, which showed a significant correlation of the CTC status and therapeutic efficacy [24]. 

To date, there is no “golden standard” for CTC enrichment and isolation. In this study, we used the technically simple, easy, cost-effective, and fast size-based filtration for CTC isolation. Other studies that used the label-free SceenCell^®^ method in comparison with the epitope-dependent CellSearch^®^ technology found higher CTC rates with the size-based techniques in head, neck, and pancreatic cancer [25]. The biologic consequence of these isolated CTCs was, however, not clear. One other study on non-small cell lung cancer (NSCLC) used the ScreenCell^®^ technique in 23 resectable patients and did not find a correlation of CTC positivity and poorer survival [26].

Since the detection of CTCs by this method is plainly by size and cell-morphology, the probability of false positive (and negative) results is higher than in methods using cell surface antigens. It is also possible that filtration does isolate a more complete but a less relevant fraction of CTCs from the blood. To reach higher specification of the cells in transit, we used epithelial cell-surface labeling for pan-cytokeratin. Additionally, we analyzed the presence of vimentin to investigate the possible presence of EMT with expression of mesenchymal markers. All cells were positive for both markers, but with diverse intensity. On the other hand, only 23% of the samples of the primary tumor had positive expression of vimentin, which supports the presumption that the CTCs undergo EMT. Since the CTCs were stained directly on the filter after removing the May-Grünwald Giemsa staining, the stain-quality was lower than the control immunofluorescence labeling done directly on the control slides. This may have led to false positive or negative results. One may also argue that CD45 staining of CTC samples is missing to exclude white blood cells as potential false positive targets. We are, however, convinced that CTC identification on bright field microscopy by experienced cytopathologists is equal or even superior to purely marker-dependent CTC definitions. In clinical oncology practice, cancer cells are cytopathologically diagnosed in fluids and smears in bright field microscopy. Additional immunostaining is only performed for characterization of the specific tumor entity. 

In summary, the present study proves the feasibility of CTC detection in non-metastatic EAC by an antigen-independent size-based filtration method. The rate of CTC-positive findings and the absolute quantity of CTCs are affected by both neoadjuvant chemotherapeutic or radiochemotherapeutic treatment and by surgery. Morphological diversity with single-cell CTCs and clusters of CTCs, but also frequent co-expression of epithelial and mesenchymal markers, were evident. The present study cannot make a statement about the prognostic significance or possible therapeutic consequences of the observed increase of CTCs after therapy or the role of the different morphological CTC subtypes. These questions are currently being investigated in a translational multicenter study conducted with the therapeutic ESOPEC trial (NCT02509286) [27]. This study compares the epithelial cell surface antigen-based method (CellSearch^®^) with the size-based filtration method (ScreenCell^®^) for the detection of CTCs in multimodal treatment of EAC with a planned sample size of 438 patients and a clinical follow-up of a minimum of three years. Over 300 patients have already been enrolled. 

## 4. Materials and Methods

### 4.1. Study Design

This study was conducted as a prospective study in the Comprehensive Cancer Center Freiburg at the University of Freiburg Medical Center, Germany and was approved by the local Ethics Committee (EK No. 190/15). All patients gave full informed consent for material and data acquisition and for all of the following experiments. Patients with histologically-proven EAC in a non-metastatic stage were consecutively enrolled in this study prior to beginning treatment of the disease. The patients had no previous medical history of cancer. The enrolled patients had locally advanced tumor disease (UICC TNM > cT2 cNx cM0) and qualified for a multimodal therapy concept. The decision for the neoadjuvant treatment (chemotherapy (PeriCTX): FLOT Protocol (5-FU/leucovorin/oxaliplatin/docetaxel) vs. chemoradiation (NeoCRT): CROSS protocol (41.4 Gy plus carboplatin/paclitaxel)) was made outside the study by the multidisciplinary tumor board. Additionally, we analyzed blood specimens of 10 healthy volunteers for the presence of CTCs.

Blood specimens were sampled at three time-points: before the start of the neoadjuvant therapy (time-point 1 (TP1)), after the neoadjuvant therapy (time-point 2 (TP2)), and after surgical resection of the tumor (time-point 3 (TP3)) (Figure 5). All patients who were oncologically and functionally resectable after neoadjuvant treatment underwent a thoracoabdominal esophagectomy (Ivor-Lewis operation). The follow-up was conducted one and two years after the operation. We investigated local recurrence, distant metastasis, and death.

### 4.2. CTC Analysis

CTC enrichment was performed by cell size-based filtration (ScreenCell^®^) using 6 mL of patient’s peripheral blood. The blood samples were processed through two ScreenCell^®^ filtration devices (Paris, France), according to the manufacturer’s instructions (3 mL blood per filtration device). CTC isolation was performed within three hours of the blood draw. The ScreenCell^®^ filtration devices are fitted with microfilters that capture the cells on small metal-rimmed filters via low-pressure vacuum filtration. After CTC-isolation, the filters were stained with a standard May-Grünwald Giemsa staining for CTC identification. Cells considered positive for CTCs were epithelioid cells with enlarged (5–20× filter pore size) irregular, hyperchromatic nuclei with well-defined cytoplasm or smaller epithelioid cells (2–5× filter pore size) with round to oval nuclei with well-defined cytoplasm in small or large clusters. CTCs were counted on each filter and the number of CTCs was divided by three to estimate the number of CTC/mL. No cut-off was chosen for CTC evaluation to maintain a complete picture. Two pathologists with one who was a very experienced cytopathologist (M.B.P) reviewed all CTC specimens. Both were blinded to the diagnosis of the patients, reviewed the cells, and made decisions on bright field without immunofluorescence staining.

The May-Grünwald-Giemsa staining was then removed from the filters positive for CTC, using phosphate-buffered saline (PBS) with 0.05% Tween 20 (20 min). For optimal epitope retrieval, a HIER (heat induced epitope retrieval) was performed. For this, the filters were boiled for 20 min using Tris/EDTA Puffer (pH 9), then washed with PBS (4×), and blocked with 2% NGS prior to immunofluorescence staining. This was performed using double immunolabeling with a pan-cytokeratin antibody (mouse, #ab7753, Abcam, Cambridge, UK) and anti-vimentin (rabbit #GTX 16700, Gene Tex, diluted 1:100, Irvine, CA, USA) applied to the filters overnight at 4 °C. Filters were then washed four times with PBS and incubated with the secondary antibodies anti-mouse Alexa Fluor 488 (A-11029, Thermo Fisher, diluted 1:1000, Waltham, MA, USA) and anti-rabbit Cy3 (A 10520, Thermo Fisher, diluted 1:1000) for one hour. Afterward, the filters were washed twice with PBS, stained with the standard nuclei-stain DAPI (= Hoechst 33342), and washed again with PBS (2×) and water (1×). We defined a very low cut-off of 1 CTC per blood specimen to classify a sample “CTC-positive.” Prior to the patient sample, staining spiking experiments were performed to develop the staining protocol. Esophageal adenocarcinoma cell lines FLO-1 and SKGT-4 were spiked into human blood samples and filtered with the ScreenCell^®^ devices.

### 4.3. Tissue Analysis

Standard histological analysis of the tumor from the operative specimen was performed, according to the UICC TNM Classification [28], including tumor size (T), degree of spread to regional lymph nodes (N), invasion into the lymphatic vessels (L), resection status (R), and the tumor regression grade, according to Becker [29]. Furthermore, immunohistochemistry with commercially-available vimentin antibodies was performed on paraffin-embedded sections of the primary tumor from each operated patient. The expression of vimentin was investigated by a pathologist blinded to the results of the CTC analysis. The intensity of the stain was evaluated by eye and scored as weak (1), moderate (2), and strong (3) labelling.

### 4.4. Statistics

Due to the sample size, only summary statistics were applied.

## 5. Conclusions

This is the first study to demonstrate CTCs in EAC after neoadjuvant therapy and surgery. It shows that detection of CTC by filtration in non-metastatic EAC is feasible. The rate of CTC positive findings and the quantity of CTCs changes in the course of multimodal neoadjuvant chemoradiation/chemotherapy and surgery. Future studies with larger sample sizes and multicenter design are currently recruiting and will give an answer regarding a question for the predictive impact of CTCs on treatment decisions for EAC. 

## Figures and Tables

**Figure 1 cancers-11-00397-f001:**
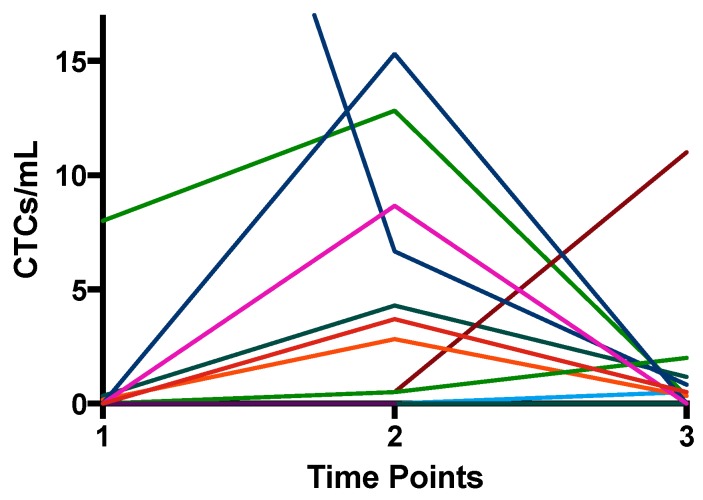
Number of CTC/mL (single CTCs + cluster CTCs) isolated by ScreenCell^®^ at the three time-points. Every line represents the CTCs of one patient over the time course.

**Figure 2 cancers-11-00397-f002:**
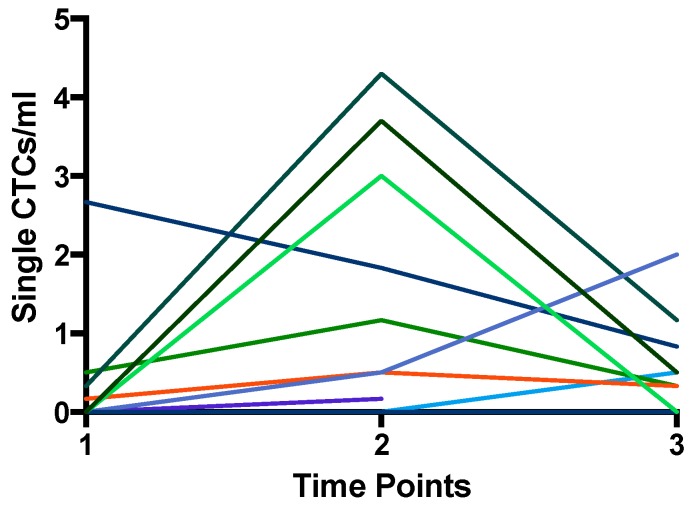
Number of single CTC/mL isolated by ScreenCell^®^ at the three time-points. Every line represents the CTCs of one patient over the time course. Notable is the increase of the CTC count after neoadjuvant treatment (TP2).

**Figure 3 cancers-11-00397-f003:**
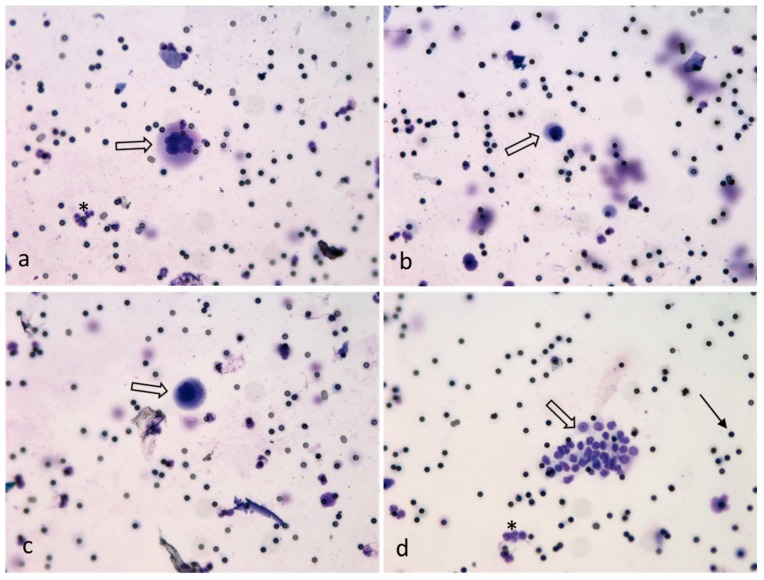
Exemplary pictures of the different morphology of single- (**a**–**c**) and cluster-CTCs (**d**) marked with white arrows. Cells isolated by ScreenCell^®^, May-Grünwald Giemsa staining (20× magnified). Filter pores (7.5 µm) marked with simple black arrow and white blood cells marked with asterisk.

**Figure 4 cancers-11-00397-f004:**
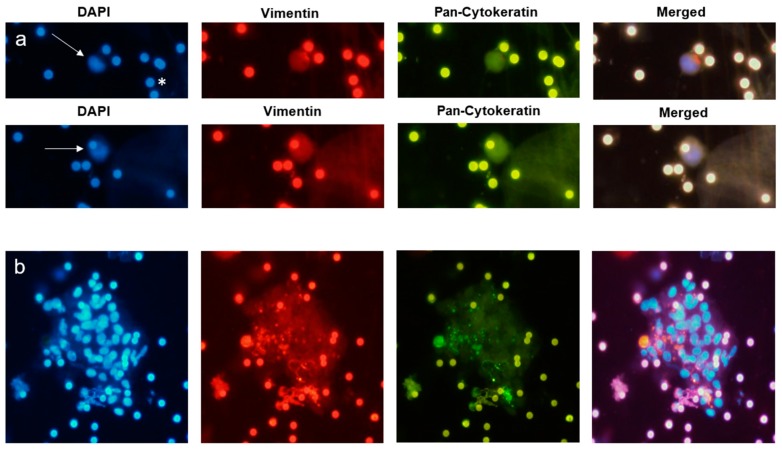
Exemplary pictures of the Immunofluorescence labelling of single CTC (**a**, marked with an arrow) and Cluster-CTC (**b**), 20× magnified. After the May-Grünwald Giemsa staining was removed, the filters were stained with 4′,6-diamidino-2-phenylindole (DAPI) Vimentin, and Pan-Cytoceratin. The pores of the filters show autofluorescence (*).

**Figure 5 cancers-11-00397-f005:**
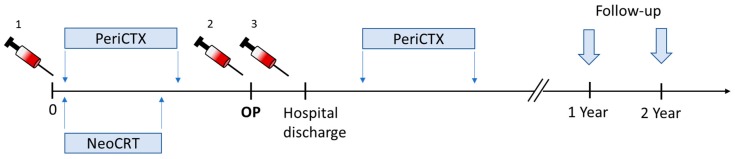
Representation of the study timeline. Blood specimens were collected before the start of the neoadjuvant treatment (PeriCTX = chemotherapy vs. NeoCRT = radio chemotherapy (TP1)), one day before the operation (TP2), and lastly before hospital discharge (TP3). The follow-up was conducted one and two years after the operation. The duration of the neoadjuvant treatment was eight weeks for the patients with chemotherapy, according the FLOT protocol, and five weeks for the patients who underwent radiochemotherapy, according to the CROSS protocol.

**Table 1 cancers-11-00397-t001:** Patient characteristics and number of circulating tumor cells (CTC) positive patients (results by ScreenCell^®^) at the three time-points.

Variables	*n*	CTC Positive (*n*)		
		TP1	TP2	TP3
All	20	6	9	8
Sex				
Male	18	2	1	1
Female	2	4	8	7
Resection				
Yes	15			
No	4			
unknown	1			
T-Stage				
ypT0	3	0	1	1
ypT1	4	2	3	3
ypT2	3	1	2	2
ypT3	4	2	2	2
ypT4	1	0	0	0
Missing	5			
N-Stage				
ypN0	7	3	5	5
ypN1	4	1	1	2
ypN2	2	1	1	1
ypN3	2	0	1	0
Missing	5			
Regression status				
complete	2	0	1	1
subtotal	6	3	4	4
partial	3	1	1	1
minimal	4	1	2	2

**Table 2 cancers-11-00397-t002:** Display of the separate patients with the individual treatment, tumor characteristics, CTC/mL, and follow-up. The CTC count is represented as single CTC/mL and CTC/mL (single CTC + cluster-CTC). Vimentin expression: 1 = weak, 2 = moderate, 3 = strong. Samples not processed due to technical difficulties or loss to follow-up marked with *.

Patient	Treatment Regimen	Resection	TP1		TP2		TP3		pTNM			Pathology	Tumor	1-Year Follow-up			2-Year Follow-up		
			CTC/mL	Single CTC/mL	CTC/mL	Single CTC/mL	CTC/mL	Single CTC/mL	T	N	M	Regression	Vimentin Expression	No Relapse	Tumor Relapse	Death	No Relapse	Tumor Relapse	Death
**1**	FLOT	yes	0	0	3.7	3.7	0.5	0.5	1	0	0	subtotal	0	1	0	0	1	0	0
**2**	FLOT	yes	0	0	8.7	3	0	0	3	3	0	minimal	?	0	1	1	/	/	/
**3**	CROSS	yes	15.3	0	0	0	0	0	3	1	0	partial	0	0	1	0	0	0	1
**4**	FLOT	yes	0	0	0	0	/*	*	4	3	1	minimal	3	0	1	1	/	/	/
**5**	FLOT	unknown	0	0	/*	/*	/*	/*	/	/	/	/	/	/	/	/	/	/	/
**6**	FLOT	yes	0	0	0	0	0	0	0	0	0	complete	1	1	0	0	1	0	0
**7**	FLOT	yes	0	0	0	0	0	0	2	1	1	subtotal	0	0	1	0	0	0	1
**8**	FLOT	yes	0	0	0.5	0.5	2	2	0	1	0	complete	0	0	1	0	0	0	1
**9**	CROSS	no	0	0	/	/	/	/	/	/	/	/	/	0	1	1	/	/	/
**10**	FLOT	no	0	0	0.17	0.17	/	/	/	/	/	/	/	0	1	1	/	/	/
**11**	FLOT	yes	0	0	0	0	0	0	2	0	0	subtotal	0	1	0	0	1	0	0
**12**	CROSS	yes	0.2	0.2	2.8	2.3	0.3	0.3	2	0	0	minimal	0	0	1	1	/	/	/
**13**	CROSS	yes	44.3	2.7	6.7	1.8	0.8	0.8	1	0	0	subtotal	2	1	0	0	1	0	0
**14**	FLOT	no	0	0	/	/	/	/	/	/	/	/	/	1	0	0	1	0	0
**15**	FLOT	yes	0	0	0.5	0.5	11	0.3	2	0	0	partial	0	1	0	0	1	0	0
**16**	FLOT	yes	0.3	0.3	4.3	4.3	1.2	1.2	3	2	0	subtotal	0	1	0	0	1	0	0
**17**	FLOT	no	0.2	0.2	/	/	/	/	/	/	/	/	/	/	/	/	/	/	/
**18**	CROSS	yes	8	0.5	12.8	1.2	0.3	0.3	1	0	0	subtotal	0	1	0	0	1	0	0
**19**	FLOT	yes	0	0	0	0	0.5	0.5	3	1	0	minimal	0	1	0	0	1	0	0
**20**	FLOT	yes	0	0	/*	/*	/*	/*	1	2	1	partial	0	0	1	1	/	/	/

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
