# Peer review of "Non-Metastatic Esophageal Adenocarcinoma: Circulating Tumor Cells in the Course of Multimodal Tumor Treatment"

_cancers, 2019, doi:10.3390/cancers11030397_

Round 1

Reviewer 1 Report

The authors evaluate a commercially available device, Screencell, for the detection of CTCs by filtration. They conclude that 1. Detection of CTC by filtration within multimodal treatment protocols of non-metastatic EAC is feasible and 2. The rate of CTC positive findings and the quantity of CTCs are affected by both neoadjuvant treatment and surgery.

Some major comments:

The conclusions/findings are not impactful. Several other literatures has explored similar conclusions, albeit not in EAC.

The method and application used for isolation are also not novel

Figure 1 legend is incomplete

Figure sequences are wrong

Numerous spelling inconsistencies e.g. PERICTX or PERCTX?

Numerous spelling errors e.g. line 63

Insufficient illustration of findings – perhaps correlation to CTC counts to other patient parameters (besides regression, TPs and resection) or genetic analysis of CTCs?

Some of the cells in Figure 4 do not resemble cancer cells due to nuclei structure

Background of immunostaining on filters is too high to evaluate the actual expression in cells

Author Response

Comment 1: The conclusions/findings are not impactful. Several other literatures has explored similar conclusions, albeit not in EAC.

Answer: Thank you very much for your time and the review of this manuscript.

Indeed we present results from a pilot study with a limited number of patients and thus with related results (no correlation with survival, no genetic analysis etc.). But as mentioned, we give the very first description of CTC in patients with resectable adenocarcinoma of the esophagus (EAC) under multimodal treatment conditions. These protocols are the modern international standard of treatment in EAC. To develop a valid statement on the prognostic and predictive validity of new biomarkers, meaningful clinical examination must be carried out in contemporary treatment protocols.  Moreover, the change of CTC counts over the course of multistep treatment has not been described anywhere for this entity. The present study is the basis for a currently recruiting biological substudy of the ESOPEC trial with over 400 patients.

Comment 2: The method and application used for isolation are also not novel

Answer: The novelty of the present study is both the application of the described isolation/detection method in the specific entity (EAC) – and the analysis of CTC in the course of very contemporary treatment protocols (FLOT/CROSS).

Indeed, the filtration method by ScreenCell itself is not novel. The goal of the study was to evaluate CTC isolation in patients with resectable EAC over the time course of multimodal treatment. Indeed, many different (and potentially easier, technically more sophisticated etc.) CTC isolation techniques are being developed; there is yet no “golden standard” for CTC isolation. Any of these “novel” techniques has to go the way we started in the present study for the described filtration method before becoming of any impact for the treatment of human patients.

I hope this emphasizes the fact that we are interested in the clinical relevance of CTC and discussed the results of purely size-based isolation more critically (Page 9). We adjusted the manuscript to make this clearer (e.g. page 7 and 9, .

Comment 3: Figure 1 legend is incomplete

Answer: Correct, we completed the figure legend.

Comment 4: Figure sequences are wrong

Answer: Correct, we adjusted the figure sequence.

Comment 4: Numerous spelling inconsistencies e.g. PERICTX or PERCTX?

Answer: Thank you we use consistent spelling in the revised manuscript.

Comment 5: Numerous spelling errors e.g. line 63

Answer: We again had a professional English language editing- all spelling mistakes should be corrected.

Comment 6: Insufficient illustration of findings – perhaps correlation to CTC counts to other patient parameters (besides regression, TPs and resection) or genetic analysis of CTCs?

Answer: This is a very good and important point. The question of genetics has been extensively discussed in our team. Unfortunately, EAC is not known to harbor specific characteristic mutations such as KRAS in pancreatic cancer (we are very familiar with KRAS). A future project is, however, to evaluate the mutational landscape of EAC CTC (and the primary tumor) to see whether or not there are specific mutations that can be recurrently found in CTC of EAC. For the moment we improved the illustration of findings: we describe in more detail the potential effect of neoadjuvant treatment on the presence and count of CTCs in single example patients (page 4, line 111-130).  

Comment 7: Some of the cells in Figure 4 do not resemble cancer cells due to nuclei structure

Answer: Thank you very much for this comment. All CTC specimens were reviewed by two pathologists, one of them a very experienced cytopathologist ( M.Pitman, MGH Boston USA).  Both reviewed the cells and made decisions on bright field. In Figure 4, we demonstrate 3 single CTCs (large lumped nuclei, large cells, high chromatin content) and 1 cluster. Within this cluster there are indeed 2 or 3 cells that appear rather benign. The other cells in the cluster are more clearly cancer cells due to the irregular nuclei structure and resemble FNA specimens of adenocarcinoma. I guess this discussion shows one more time that we are far away from understanding the whole story of CTCs- and of cancer dissemination in general. We added a more detailed evaluation section to the method part of the paper (page 10, line 374ff) .

Comment 8: Background of immunostaining on filters is too high to evaluate the actual expression in cells

Answer: Thank you very much for this comment. We were not sure whether or not we wanted to include the immunostaining in the manuscript due to the weak images. The filter pores give an extensive background staining. The staining of “real” circulating tumor cells on ScreenCell devices is indeed challenging. In all CTC positive patient samples we found some expression of vimentin or cytokeratin. Spiked EAC Cells are much easier to stain. We point that out clearly in the results (page 6, line 161 ff) and discuss the limitations in the Discussion (page 8 258 f) . If you prefer the figure to be removed since “intensity scoring” was not possible, it is an option.  

Reviewer 2 Report

The manuscript describes the use of filtration based CTC enrichment technology for esophageal carcinoma during therapy. This manuscript has been well written and has a good study design. However a number of concerns need to be addressed:

Only 1 CTC is determined as positive - what is the sensitivity/specificity of this assay? Do the authors find CTC-like cells in normal healthy volunteers? Using other technologies, cut off's have been established for CTCs where some larger cells have been found in healthy volunteers. 

2. In figure 1, there is no image of CD45 staining - how do we know whether these are white blood cells?

3. Can the authors compare the screencell findings to other studies using screencell technology for ctc capture (eg. Kulasinghe et al., Oncotarget Impact of label free technologies in head and neck cancer circulating tumour cells; Chudasama et al., Anticancer research 2017).

4. Did the authors find any clogging issues with the Screencell filters? These systems are known for clogging during the filtration process?

5. Were the ctc clusters that were identified pure ctc clusters or include normal white blood cells as carriers? can the authors comment on this? There are some recent publications on ctc clusters (Aceto et al., 2019 Nature; Kulasinghe et al.,2019 Cancers)

Author Response

The manuscript describes the use of filtration based CTC enrichment technology for esophageal carcinoma during therapy. This manuscript has been well written and has a good study design. However a number of concerns need to be addressed:

Comment 1: Only 1 CTC is determined as positive - what is the sensitivity/specificity of this assay? Do the authors find CTC-like cells in normal healthy volunteers? Using other technologies, cut off's have been established for CTCs where some larger cells have been found in healthy volunteers. 

Answer: Thank you very much for your time and the review of this manuscript.

These are very good questions. Is it wise to set a cutoff? The problem with the cutoff is the fact that you tolerate “CTC-like cells” in healthy volunteers. Why should there be such cells? And why not call the sample “(false) positive”? We have plenty of experience with the ScreenCell device in pancreatic cancer and tested many healthy volunteers  -  we do not find typical CTC in healthy volunteers. Sometimes we see “larger naked nuclei” in healthy volunteers, but no actual CTC. We thus decided the lowest cutoff level possible: 1 CTC per sample. In our view, every cell counts – especially in patients with resectable cancer and thus lower disease burden. In table 2, we extensively display the number of CTCs per patient over the course of the treatment. Future analysis from our group will show if specific cutoff levels are relevant as predictive markers. It might very well be possible that patients with >10 CTC/ml have an inferior outcome, and lower CTC burden is irrelevant. The currently recruiting biological substudy of the ESOPEC trial will hopefully shed light on some of these questions.

Regarding Specificity and Sensitivity: if we use our latest 10 healthy volunteers (none positive) and we see 6 out of 20 patients are positive at diagnosis, we have a sensitivity of 0.3 and a specificity of 1. The positive predictive value is 1, the negative predictive value 0.417.

 Since the CTC isolation was however not used as a screening tool in this setting of 20 patients, we did not include it in the manuscript. 

Comment 2: In figure 1, there is no image of CD45 staining - how do we know whether these are white blood cells?

Answer: This is also a very good point. We have only a 3-channel immunofluorescence microscope. We thus decided to use DAPI, CK and vimentin to have as much information on each CTC. Additionally, the CD45 stain was often weak and problematic. We however agree that this has to be addressed in the future. Two pathologists, one of them a very experienced cytopathologist, reviewed the cells and made decisions on bright field.

Additionally, cells positive for cytokeratin should also not be blood components. And since almost every CTC showed a cytokeratin stain, we are confident that these are tumor cells in transit. Finally, we indeed do not know what these cells are (even if they were CD45 negative).

Comment 3: Can the authors compare the screencell findings to other studies using screencell technology for ctc capture (eg. Kulasinghe et al., Oncotarget Impact of label free technologies in head and neck cancer circulating tumour cells; Chudasama et al., Anticancer research 2017).

Answer: Yes, this is an important point. We added the key findings to the discussion of the manuscript.

Comment 4: Did the authors find any clogging issues with the Screencell filters? These systems are known for clogging during the filtration process?

Answer: Thank you for this very interesting question. Indeed we sometimes had clogging issues in patients with pancreatic cancer  - mostly with high tumor burden.

In this series we had clogging issues in 1 specimen (we added this information to the manuscript p3, line 95).

It is important to avoid any kind of clogging immediately after the blood draw by inverting the tube several times. Additionally, we found that clogging can be an issue if there are “too many CTCs” in one specimen – and sometimes patients seem to have “sticky blood”. In these cases, we sometimes added some more PBS to the top unit and used a second collection tube to maintain the vacuum. If that was still impossible and the filter was “stuck”, we used a second filter and pipetted the supernatant to a second filter unit. This is clearly not the idea of the system, and does not happen frequently (1 or 2 in 50 specimens), but we avoid cell and sample loss with this rescue technique.

Comment 5: Were the ctc clusters that were identified pure ctc clusters or include normal white blood cells as carriers? can the authors comment on this? There are some recent publications on ctc clusters (Aceto et al., 2019 Nature; Kulasinghe et al.,2019 Cancers)

Answer: Thank you for this important point. The clusters were not extensively reviewed for white blood cell components, but in view of the recent publications we went back and evaluated each cluster. Most clusters were pure CTC clusters, only a few clusters showed single eosinophilic granulocytes on or close to the cluster. We added a section about CTC Cluster (page 7, 228 ff).

Round 2

Reviewer 1 Report

The authors presented a study which is interesting in the aspects of CTC detection in patients with resectable adenocarcinoma of the esophagus (EAC). Studies will provide new insights for this particular cancer type. However, some controls are still missing that confirms the identification of CTCs.

1.    The authors mentioned that all cells were positive for both markers, but with diverse intensity. Did the authors do control with cell lines filtered by Screen cell? This is a concern regarding the integrity of the cells after filtration that may have led to high background staining

2.    CD45 staining seems to be missing, and negative expression of CD45 is usually one of the key factors for defining CTCs

Author Response

The authors presented a study which is interesting in the aspects of CTC detection in patients with resectable adenocarcinoma of the esophagus (EAC). Studies will provide new insights for this particular cancer type. However, some controls are still missing that confirms the identification of CTCs.

COMMENT 1: The authors mentioned that all cells were positive for both markers, but with diverse intensity. Did the authors do control with cell lines filtered by Screen cell? This is a concern regarding the integrity of the cells after filtration that may have led to high background staining.

ANSWER 1: Thank you very much for your time and the review of this manuscript. The immunofluorescence (IF) staining of CTC is challenging.  We indeed controlled the staining behavior of two different cell lines (SK-GT-4 and FLO-1) on smears but also after spiking into whole blood and filtration with the ScreenCell devices. We added a short description to the method section (page 11, line 361-364).  IF was best on smears but was also good in the spiking experiments. In our experience cell lines stain more intense on immunofluorescence than the actual cytophathologically identified CTC (not only in esophageal cancer but also in pancreatic cancer, data not published). Differences have been described between primary tumor tissue and derived cell lines (e.g. Ertel A. et al. Pathway-specific differences between tumor cell lines and normal and tumor tissue cells Mol Cancer. 2006; 5: 55.)- This is a potential explanation apart from the cell integrity after filtration- that is indeed always a concern. If requested, we can include a supplementary figure with images of the spiking experiments.

COMMENT 2:  CD45 staining seems to be missing, and negative expression of CD45 is usually one of the key factors for defining CTCs

ANSWER 2: This is also a very good point. We have only a 3-channel immunofluorescence microscope. We thus decided to use DAPI, CK and vimentin to characterize each CTC as specific as possible. Additionally, the CD45 stain was often weak and problematic. We however agree that this has to be addressed in the future. Two pathologists, one of them a very experienced specialized cytopathologist, reviewed the cells and made the decisions on bright field. This is emphasized in the methods section line 341-346. Many studies using the CellSearch device (and others) mainly rely on surface marker to decide about “malignancy” of a cell. We are convinced that bright field is more important for diagnosis than a marker specific panel without bright field. A combination of both – including CD45 is ideal.

Moreover, cells positive for cytokeratin as shown in our study should not be blood components. And since most CTC showed a cytokeratin stain, we are confident that these are tumor cells in transit. Finally, we indeed do not know what these cells are (even if they were CD45 negative). Unfortunately we cannot go back and re-stain our samples.

In clinical oncology practice, cancer can be cytopathologically diagnosed in fluids and smears. The cytospins or smears are primarily stained for bright field analysis. After a “malignant” diagnosis cells are stained with immunofluorescence for more detailed characterization. This is common practice in e.g. pleural or peritoneal effusions with tumor cells originating from solid cancers and supports the idea of CTC diagnosis by cytopathology.

We added an appropriate section to the discussion to address this problem. (page 9, line 276-282)